# Expansion Microscopy Reveals *Plasmodium falciparum* Blood-Stage Parasites Undergo Anaphase with A Chromatin Bridge in the Absence of Mini-Chromosome Maintenance Complex Binding Protein

**DOI:** 10.3390/microorganisms9112306

**Published:** 2021-11-06

**Authors:** Benjamin Liffner, Sabrina Absalon

**Affiliations:** Department of Pharmacology and Toxicology, Indiana University School of Medicine, Indianapolis, IN 46202, USA; Bliffner@iu.edu

**Keywords:** *Plasmodium*, malaria, apicomplexa, nuclear division, closed mitosis, expansion microscopy, nuclear envelope, cell division

## Abstract

The malaria parasite *Plasmodium* *falciparum* undergoes closed mitosis, which occurs within an intact nuclear envelope, and differs significantly from its human host. Mitosis is underpinned by the dynamics of microtubules and the nuclear envelope. To date, our ability to study *P. falciparum* mitosis by microscopy has been hindered by the small size of the *P. falciparum* nuclei. Ultrastructure expansion microscopy (U-ExM) has recently been developed for *P. falciparum*, allowing the visualization of mitosis at the individual nucleus level. Using U-ExM, three intranuclear microtubule structures are observed: hemispindles, mitotic spindles, and interpolar spindles. A previous study demonstrated that the mini-chromosome maintenance complex binding-protein (MCMBP) depletion caused abnormal nuclear morphology and microtubule defects. To investigate the role of microtubules following MCMBP depletion and study the nuclear envelope in these parasites, we developed the first nuclear stain enabled by U-ExM in *P. falciparum**.* MCMBP-deficient parasites show aberrant hemispindles and mitotic spindles. Moreover, anaphase chromatin bridges and individual nuclei containing multiple microtubule structures were observed following MCMBP knockdown. Collectively, this study refines our understanding of MCMBP-deficient parasites and highlights the utility of U-ExM coupled with a nuclear envelope stain for studying mitosis in *P. falciparum*.

## 1. Introduction

Malaria is estimated to cause over 400,000 deaths annually. These deaths are predominantly in young children, and are caused by the unicellular protozoan pathogen *Plasmodium falciparum* [1]. Resistance against frontline antimalarials has emerged in many parts of the globe and is spreading [2,3,4,5,6]. Moreover, there is no highly effective vaccine against malaria, highlighting the need to develop new therapeutic interventions for ongoing and future control of this disease. One therapeutic strategy is the drug inhibition of DNA/RNA replication, and/or cell division, a method that is commonly used for the control of bacterial [7] and viral diseases [8], along with many types of cancer [9]. The distinctive division method of *P. falciparum* compared to its human host makes this an attractive strategy for *P. falciparum* drug design, yet no current antimalarials directly target DNA replication or cell division [10], highlighting the need for further investigation into this pathway.

*Plasmodium* parasites undergo cell division by a process known as schizogony, whereby a singly nucleated parasite undergoes repeated rounds of DNA replication and mitosis, within a shared cytoplasm, followed by a single cytokinetic event that results in the formation of 16–32 daughter parasites [11,12,13]. Throughout division, *Plasmodium* undergoes closed mitosis, where the nuclear envelope remains intact, as opposed to the open mitosis of its human host [14]. Nuclear division during schizogony is orchestrated by a set of intranuclear microtubule structures, which appear to be unique to *Plasmodium* [15,16]. The first observed microtubule structure during the blood-stage of the lifecycle is the hemispindle, a collection of around five microtubule branches that extend from a single microtubule organizing center (MTOC) throughout the nucleus [15,16,17]. It has been shown that during the first mitosis, the hemispindle retracts, the MTOC duplicates, and the mitotic spindle is formed [16]. Mitotic spindles consist of short microtubules that extend from two opposing MTOCs and connect to the kinetochore, in preparation for chromatid separation into the daughter nuclei [15,16,18,19,20,21]. Following the chromatid separation, two distinct and distant DNA masses are observed, each containing their own MTOC, but connected by extended microtubules that are attached to both MTOCs [16]. The nomenclature around this microtubule structure is inconsistent, having previously been named astral microtubules [14], complete hemispindles [15], and extended spindles [16], but in this study, they will be referred to as interpolar spindles, as their defining feature is connecting two distant MTOCs. Following nuclear segregation, the interpolar spindle retracts and the two daughter nuclei undergo nuclei fission before reforming the hemispindle [16].

Mitosis in *Plasmodium* has been poorly studied historically. This is due to the small size of *P. falciparum* nuclei (~1 µm diameter), which largely prevents the visualization of events occurring at a single-nucleus level by conventional microscopy techniques. Technological advancements that overcome the resolution hurdle, along with recent work on other organisms that undergo closed mitosis, have reignited interest in closed mitosis broadly, and of *Plasmodium* mitosis specifically. The application of Ultrastructural-expansion microscopy (U-ExM), a sample preparation method that isotropically expands the sample ~4.5×, to *P. falciparum* [22] has radically enhanced our ability to study the biology occurring inside individual nuclei. Indeed, U-ExM has already been used to observe microtubules in asexual blood-stages, ookinetes and gametocytes [16,22,23]. Moreover, U-ExM has been used to differentiate the intranuclear microtubules of *P.* falciparum, providing significant insight into the physiology of microtubule dynamics in the process [16,22]. Outside of *Plasmodium*, a recent study on the genetically tractable and much larger fission yeast that undergoes closed mitosis, *Schizosaccharamyces pombe*, provided mechanistic insight into this fascinating process, with a particular focus on the relationship between closed mitosis and nuclear envelope dynamics [24]. With the application of U-ExM to *P. falciparum*, we can now begin to study the closed mitosis of *Plasmodium* with a resolving power similar to fission yeast, the most well studied model organism for closed mitosis. Notably, however, the absence of a uniform marker for the nuclear envelope in *Plasmodium* largely precludes the understanding of the role of the nuclear envelope in closed mitosis.

Studies on *Plasmodium* microtubules historically have been largely descriptive, and relatively few individual proteins are known to influence microtubule dynamics. One protein that has previously been shown to be involved in *P. falciparum* microtubule organization is the mini-chromosome maintenance binding protein (MCMBP) [25]. In other organisms, MCMBP is directly involved in DNA replication, with disruption or alteration in its expression leading to defects in DNA replication and nuclear morphology, and leading to MTOC amplification [26,27,28]. In a recent study, the authors generated a transgenic parasite line for inducible knockdown where the destabilization domain system was incorporated into MCMBP (MCMBP^HADD^) [25]. Using this system, the addition of a molecule called Shield-1 (Shld1) results in a wildtype expression of MCMBP, while the absence of Shld1 results in specific MCMBP knockdown. MCMBP-deficient parasites were able to undergo DNA replication, but showed severe microtubule defects and aneuploidy attributed to defective DNA segregation [25]. Given then canonical role of MCMBP in DNA replication, it is likely that the microtubule defects observed are downstream, rather than direct, effects of MCMBP depletion. Importantly, while this work was performed with state-of-the-art Airyscan microscopy at the time of publication, they were unable to visualize several important facets of mitosis and microtubules in MCMBP-deficient parasites due to the resolution limit of this technique. Notably, the limited resolution meant it could not be determined whether aberrant microtubules represented hemispindles or interpolar spindles. Additionally, the absence of a nuclear envelope marker meant that it could not be determined whether microtubule structures were present in the shared or separate nuclei.

Here, we use U-ExM to study the spatial organization of intranuclear microtubules and nuclear division in the context of MCMBP-deficient parasites. While the effects of MCMBP knockdown on microtubules are likely downstream of MCMBP function, *Plasmodium* continues DNA replication even with the presence of these defects due to a lack of canonical cell cycle checkpoints [29]. This provides the unique ability to study the ongoing impact of both MCMBP knockdown and microtubule dysregulation in parasite mitosis. Additionally, we develop the first U-ExM compatible nuclear envelope stain. We use U-ExM to show that MCMBP-deficient parasites display defective hemispindles and mitotic spindles. Additionally, we couple U-ExM and nuclear envelope staining to show that MCMBP-deficient parasites form anaphase chromatin bridges, which leads to aneuploidy and the presence of multiple microtubule structures contained within the same nuclear envelope.

## 2. Materials and Methods

### 2.1. Plasmodium Falciparum Culture

All parasites used in this study were the previously generated transgenic parasite line 3D7-PfMCMBP^3HADD^ [25]. Routine culture parasites were grown in O^+^ human red blood cells at a hematocrit of 4% in RPMI-1640 supplemented with 50 mg/L of hypoxanthine, 25 mM of HEPES, 0.5% *w*/*v* Albumax II, and incubated in a mixture of 1% O_2_, 5% CO_2_, and 94% N_2_ as previously described [30]. Additionally, to prevent the degradation of MCMBP through the destabilization domain system, parasites were cultured in the presence of 250 nM Shield-1 as previously described [25,31].

Parasites were tightly synchronized using a combination of Percoll concentration and sorbitol lysis. Briefly, schizont-stage cultures were resuspended in 60% Percoll and separated from uninfected red blood cells by centrifugation as previously described [32]. Concentrated schizonts were then allowed to reinvade new red blood cells for ~3 h under normal culture conditions. Following reinvasion, parasite cultures were resuspended in 5% *w*/*v* D-sorbitol to selectively lyse schizonts that had not reinvaded, as previously described [33].

For assays requiring the knockdown of MCMBP, and therefore the washout of Shield-1, synchronized schizont-stage cultures (~44–46 h.p.i.) were separated from uninfected red blood cells using QuadroMACS^®^ magnet activated cell sorting [34]. Parasites were allowed to reinvade in the absence of Shield-1 for ~3 h before being synchronized with sorbitol. As it has been previously shown that the knockdown of MCMBP delays parasite growth by approximately three hours [25], parasites grown in the absence of Shield-1 were always collected three hours after those grown in the presence of Shield-1.

For assays where segmented schizonts were harvested, schizont-stage cultures (~44 h.p.i.) were treated with the schizont egress inhibitor [35] trans-Epoxysuccinyl-L-leucylamido(4-guanidino)butane (E64) at 10 µM for ~5 h.

### 2.2. Immunofluorescence Assays

Immunofluorescence assays in this study were adapted from previously published protocols [17,25]. Briefly, for immunofluorescence assays of unexpanded parasites, ~1 mL of parasite culture at 2% hematocrit was centrifuged at 2000 rpm in a benchtop centrifuge, the supernatant removed, and the culture resuspended in 4 % *w*/*v* paraformaldehyde-PBS before incubating at room temperature for 10 min. Fixed parasite cultures were again centrifuged at 2000 rpm, with the fixative removed and ~3 µL of packed red blood cell pellet smeared onto a glass slide and dried. The dried smears were washed three times in PBS before being permeabilized with 0.1% *v*/*v* Triton-X-100 for 10 min at room temperature. Following permabilization, smears were washed three times in PBS and blocked in 3% *w*/*v* bovine serum albumin in PBS for 60 min at room temperature. After blocking, smears were incubated with primary antibodies diluted in blocking solution for one hour at room temperature, followed by three washes in PBS. The slides were then incubated with secondary antibodies and NHS ester diluted in PBS for one hour at room temperature. Following antibody staining, the slides were washed three times in PBS, dried, mounted with ProLong™ Glass with NucBlue™ (ThermoFisher Cat. No. P36981, Waltham, MA, USA) and a #1.5 coverslip placed on top of the smear.

### 2.3. Ultrastructure Expansion Microscopy

Ultrastructure Expansion Microscopy (U-ExM) was performed as previously described [16,22,36], with significant modification.

Coverslips (Fisher Cat. No. NC1129240) that were 12 mm round were treated with poly-d-lysine for 1 h at 37 °C, washed twice with MilliQ water, and placed in the wells of a 12-well plate. Parasite cultures were set to 0.5% hematocrit, and 1 mL of parasite culture was allowed to settle onto the coverslip for 15 min at 37 °C. Culture supernatants were removed, and cultures were fixed with 1 mL of 4% *w*/*v* PFA/PBS for 15 min at 37 °C. Following fixation, coverslips were washed three times with PBS pre-warmed to 37 °C before being treated with 1 mL of 1.4 % *v*/*v* formaldehyde/2% *v*/*v* acrylamide (FA/AA) in PBS. After the addition of the FA/AA solution, the 12 well-plate was parafilmed shut and left to incubate at 37 °C overnight.

A monomer solution (19% *w*/*w* sodium acrylate (Sigma Cat. No. 408220), 10% *v*/*v* acrylamide (Sigma Cat. No. A4058, St. Louis, MO, USA), 2% *v*/*v* N,N’-methyllenebisacrylamide (Sigma Cat. No. M1533) in PBS) was made in 1 mL batches on Day 1 and stored as 90 µL aliquots at −20 °C overnight.

Aliquots of 10% *v*/*v* tetraethylenediamine (TEMED; ThermoFisher Cat. No. 17919) and 10% *w*/*v* ammonium persulfate (APS; ThermoFisher Cat. No. 17874) were thawed on ice, while a humidity chamber containing parafilm was stored at −20 °C before also being placed on ice. The FA/AA solution was removed, coverslips were washed once with PBS, dried, and placed cell-side up on the parafilm in the humidity chamber. Subsequently, 5 µL of both TEMED and APS were added per 90 µL of monomer solution, which was briefly vortexed, and 35 µL was pipetted onto the parafilm before the coverslip was placed cell-side down onto the monomer solution. The gels were then incubated at 37 °C for 1 h before being transferred into the wells of a 6-well plate filled with the denaturation buffer for 15 min at room temperature (200 mM sodium dodecyl sulfate (SDS), 200 mM NaCl, 50 mM Tris, pH 9). The gels were then separated from coverslips and transferred into Eppendorf tubes containing the denaturation buffer and denatured at 95 °C for 90 min. The denatured gels were transferred into 10 cm Petri dishes filled with 25 mL MilliQ water and placed on a platform shaker for 30 min, with the water replaced twice, each for a further 30 min. After the first expansion in water, the expanded gels were shrunk by adding 25 mL to PBS washes each for 15 min. The shrunken gels were placed into the wells of a 6-well plate filled with a blocking buffer (3% BSA-PBS) and blocked for 1 h at room temperature on a platform shaker. After blocking, primary antibodies were prepared in 1 mL of blocking buffer and gels were incubated with a primary antibody overnight at room temperature on a platform shaker.

The gels were washed three times in 0.5% *v*/*v* PBS-Tween 20 (PBS-T), each for 10 min, before being incubated with 1 mL of secondary antibodies, NHS ester and/or nuclear stain diluted in PBS for 2.5 h at room temperature on a platform shaker. Following secondary incubation, the gels were washed three times in PBS-T. The stained gels were then transferred back to 10 cm Petri dishes and underwent a second round of expansion with three 30 min washes in 25 mL MilliQ water.

The diameter of the fully expanded gels was measured using a tape measure, and the expansion factor was determined by dividing the expanded gel size (in mm) by the initial coverslip size (12 mm). The gel diameter and expansion factor for all gels prepared in this study can be found Appendix A.

For the gels stained with BODIPY TR Ceramide (BODIPY TRc), sections of the expanded gel were cut and placed into the wells of a 6-well plate containing 1 mL 2 µM BODIPY TRc in MilliQ and incubated on a platform shaker overnight.

To prepare gels for imaging, small sections were cut from the larger gel and gently dried before being placed into 35 mm #1.5 coverslip bottomed imaging dishes (Cellvis; Fisher Cat. No. NC0409658) that had been pre-coated with poly-d-lysine.

### 2.4. Stains and Antibodies

The following primary antibodies were used in this study: Mouse IgG1 anti-alpha Tubulin Clone B-5-1-2; ThermoFisher cat. No. 32-2500 (1:1000 unexpanded samples, 1:500 U-ExM samples), Mouse IgG2a anti-Centrin-1 Clone 20H5; EMD Millipore cat. No. 04-1624 (1:100 U-ExM samples), Rabbit polyclonal anti-PfBiP; this was generously provided by Dr. Jeff Dvorin (1:500 U-ExM samples).

The following secondary antibodies were used in this study: Goat anti-mouse IgG Alexa Fluor 488 Superclonal™; ThermoFisher Cat. No. A28175 (1:1000 unexpanded samples, 1:500 U-ExM samples), Goat anti-mouse IgG2a Alexa Fluor 488 Cross-Adsorbed; ThermoFisher Cat. No. A21131 (1:500 U-ExM samples), Goat anti-mouse IgG1 Alexa Fluor 594 Cross-Adsorbed; ThermoFisher Cat. No. A21125 (1:500 U-ExM samples), Goat anti-mouse IgG1 Alexa Fluor 647 Cross-Adsorbed; ThermoFisher Cat. No. A21240 (1:500 U-ExM samples), Goat anti-rabbit Alexa Fluor 488 Highly Cross-Adsorbed; ThermoFisher Cat. No. A11034 (1:500 U-ExM samples).

The following stains were used in this study: NucBlue™/Hoechst 33,342 (in ProLong Glass™ mountant), DAPI (2 µg/mL U-ExM samples), NHS Ester Atto 594 in DMSO; Sigma Cat. No. 08,741 (10 µg/mL unexpanded samples. 10 µg/mL U-ExM samples), NHS Ester Alexa Fluor 405 in DMSO; ThermoFisher Cat. No. A30000 (8 µg/mL U-ExM samples), DRAQ5™; ThermoFisher Cat. No. 62,251 (20 µM U-ExM samples), SYTOX™ Deep Red; ThermoFisher Cat. No. S11381 (1 µM U-ExM samples), BODIPY TR Ceramide in DMSO; ThermoFisher Cat. No. D7540 (2 µM U-ExM samples).

A comprehensive list of all primary antibodies, secondary antibodies, and stains used for each of the images presented in this study can be found in Appendix A.

### 2.5. Image Acquisition

All microscopy presented in this study was performed on a Zeiss LSM800 AxioObserver microscope that had an Airyscan detector. Additionally, all images were acquired using a 63× Plan-Apochromat (NA 1.4) objective lens. All images presented in this study were acquired as Z-stacks with an XY pixel size of 0.035 µm and a Z-step size of 0.15 µm. All images then underwent Airyscan processing using ZEN Blue (Version 3.1, Zeiss, Oberkochen, Germany).

### 2.6. Image Analysis

All image analysis performed in this study used ZEN Blue (Version 3.1). All measurements of length were made using the “profile” function of ZEN Blue.

To measure hemispindle branch length, maximum intensity projections were made of Airyscan-processed images. Hemispindle branches were first counted and then measured from the edge of the tubulin staining closest to the MTOC (visible on the NHS Ester channel) to the edge of the tubulin staining furthest away from the MTOC. The nuclei that contained both hemispindles and interpolar spindles were excluded from this analysis. In nuclei that contained multiple MTOCs, it could not always be determined which MTOC each branch was coming from, and so these nuclei were excluded from this analysis.

To measure mitotic spindle length, maximum intensity projections were made of Airyscan-processed images. Mitotic spindle size was measured as the greatest distance between the edge of the tubulin staining that was adjacent to each of the two MTOCs.

### 2.7. Statistical Analyses

This study reports both the measured distances of mitotic and hemispindles, and the actual estimated distances in unexpanded parasites. To estimate actual distances, the mean expansion factor of all gels used in this study was determined (4.3×; Appendix A). All actual distances were then divided by this mean expansion factor to get the actual estimated distances reported in this study.

All graphs and statistical analyses in this study were performed and generated using GraphPad PRISM (Version 9.0.0, GraphPad Software, San Diego, CA, USA). All values of statistical significance in this study were determined using an unpaired, two-tailed *t*-test.

## 3. Results

### 3.1. Ultrastructure Expansion Microscopy (U-ExM) Significantly Enhances Visualization of Microtubule Structures in P. falciparum

To validate the utility of U-ExM for visualizing microtubules in *P. falciparum*, we first confirmed that we could visualize all previously identified microtubule structures (hemispindle, mitotic spindle, interpolar spindle, subpellicular microtubules) in unexpanded parasites (Figure 1a). Additionally, we incorporated a general protein stain (N- hydroxysuccinimide (NHS) ester). In unexpanded parasites, NHS ester staining did not produce a staining pattern that obviously represented a particular organelle. Despite no obvious demarcation of organelles, NHS ester staining appeared slightly denser in the chromatin-free region of the nucleus (Figure 1a), which has previously been shown to contain the microtubule organizing center (MTOC) [16]. Additionally, in segmented schizonts, NHS ester staining appeared denser at the apical tip of merozoites, likely corresponding to the merozoite secretory organelles rhoptries, micronemes, or dense granules (Figure 1a).

Despite its unclear staining in unexpanded parasites, U-ExM parasites stained with NHS ester allowed the identification of many intracellular structures that were not recognizable in unexpanded parasites. Through differences in staining intensity, the location of the red blood cell (RBC) membrane, parasite vacuole membrane (PVM), and parasite plasma membrane (PPM) could all be inferred (Figure 1b,c). Prior to segmentation, the MTOC can be clearly identified based on NHS staining, and it roughly adopts a ‘bell-shape’ with the most intense NHS ester staining at the top of the bell, and the least at the bottom. Additionally, comparing NHS ester staining and DNA staining it can show that there is a density of NHS ester on the nuclear side of the MTOC that does not contain chromatin; this likely represents a recently identified chromatin-free nuclear compartment adjacent to the MTOC [16]. In fully segmented schizonts, the MTOC is no longer visible by NHS ester staining; we instead see the characteristic double-club-shaped rhoptries stained prominently (Figure 1b,c). At the apex of the rhoptry neck, a ring structure can be observed (Figure 1b,c), which we inferred to be the apical polar rings based on its similar appearance to the apical polar rings in electron microscopy studies [37]. It is not clear if what we observe by NHS ester staining represents apical polar ring 1, apical polar ring 2, or both. At the basal end of the parasite, we observed another ring by NHS ester staining that is likely the basal complex (Figure 1b,c) based on its similarity to the basal complex as identified by FIB-SEM [12]. By combining NHS Ester and tubulin staining, we observed that subpellicular microtubules extend from the apical polar rings, along the length of the merozoite, and end at the basal complex (Figure 1b,c). While previously published models of merozoites have speculated on this organization previously [38], to the best of our knowledge, this is the first time the apical polar rings, subpellicular microtubules, and basal complex have been observed in the same merozoite.

All microtubule structures were also observed following U-ExM, but could be observed in far greater detail with less confounding complexity from neighboring nuclei (Figure 1b). Notably, all the branches of a hemispindle could be readily differentiated, including many small branches that previously would have been below the limit of detection (Figure 1b and Figure 2a,b) (Appendix A). In mitotic spindles, both sides of the spindle, connected to either MTOC, could be differentiated and the individual branches that would connect to the kinetochore during mitosis could be observed (Figure 1b). Interpolar spindles were observed connecting two distant MTOCs (Figure 1b). Additionally, these interpolar spindles were found alongside microtubules that extended most of the way to the other MTOC, but not completely, and branches that resembled those in a hemispindle (Figure 1b). In merozoites from segmented schizonts, subpellicular microtubules were observed, with typically 2–4 individual microtubules in each merozoite (Figure 1b,c). Collectively, this shows that U-ExM, coupled with NHS ester staining, can be used to visualize *P. falciparum* microtubules at a single-nucleus level.

MCMBP^HADD^ parasites cultured in the presence of Shld1 were imaged using super-resolution Airyscan microscopy after being prepared for regular immunofluorescence assay (a), or U-ExM (b). All parasites were stained with a nuclear stain (Hoechst, DAPI, DRAQ5, or SYBR in cyan), anti-tubulin (in magenta), and a protein stain (N-hydroxysuccinimide (NHS) Ester in greyscale). All previously identified blood-stage microtubule structures (hemispindle, mitotic spindle, interpolar spindle, and subpellicular microtubules) were observed by both IFA and U-ExM. Images in (a) represent a single z-slice from a z-stack image, while images in (b) are maximum-intensity projections. Slice-by-slice videos of images in 1b are found in Videos S1–S4. Scale bars as labelled in each image are solid bars = XY scale, and dashed bar = combined depth of slices used for Z-projection. An expanded and annotated view of the NHS Ester channel from the mitotic spindle and subpellicular microtubule images from (b) along with a schematic interpretation of these images is found in (c). The arrowheads point to NHS staining of interest. The colors in schematic are black = dense NHS Ester staining, grey = light NHS Ester staining, blue = DNA, and purple = microtubules. RBC = red blood cell membrane, PVM = parasitophorous vacuole membrane, PPM = parasite plasma membrane, MTOC = microtubule organizing center, MTs = microtubules, APRs = apical polar rings, RN = rhoptry neck, RB = rhoptry bulb, BC = basal complex, Rh = rhoptry, and SPMTs = subpellicular microtubules.

### 3.2. MCMBP-Deficient Parasites Display Aberrant Hemispindles and Mitotic Spindles

It has previously been observed that MCMBP-deficient parasites display microtubule defects [25], but the resolution limit of conventional light microscopy prevented exploration of the nature of these defects. Given that each of the microtubule structures could be distinguished from each other using U-ExM (Figure 1b), we used this technique to study microtubule formation in MCMBP-deficient parasites.

MCMBP^HADD^ parasites, either in the presence or absence of Shld1, were stained with antibodies against tubulin and centrin, a nuclear stain, and the NHS ester. All images were acquired using Airyscan microscopy after U-ExM. In both hemispindles and mitotic spindles, centrin staining colocalized with the previously described ‘bell-shape’ of the MTOC observed on NHS ester staining (Figure 2a,e) (Appendix A). Notably, however, centrin staining did not colocalize with the entirety of the MTOC, with centrin foci contained within a small portion of the whole MTOC. By comparing with the nuclear stain, it could be seen that centrin foci localized toward the cytoplasmic side of the MTOC structure, suggesting that *P. falciparum* may compartmentalize subsets of proteins inside the MTOC.

In MCMBP-deficient parasites, MTOC staining often appeared aberrant with misplaced centrin foci (Figure 2e) (Appendix A). However, these defects were not consistent or easily quantifiable by regular microscopy measurement techniques. The nature of these defects is unclear, but suggest that MCMBP knockdown may alter the formation or integrity of the MTOC.

In the presence of Shld1 (Figure 2a), hemispindle branches were on average 732 nm in length (±498 nm SD) (Figure 2b), with the longest branch in each hemispindle being 1260 nm (±512 nm SD) (Figure 2c), and each hemispindle containing five branches (±2.2 SD) (Figure 2d). In the absence of Shld1 (Figure 2a), hemispindle branches were on average 31.4% longer (1067 nm ± 815 nm SD) (Figure 2b), with the longest branch in each hemispindle being 32.5% longer (1866 nm ± 936 nm SD) (Figure 2c), and each hemispindle containing 18.3% more branches (6.2 branches ± 2.8 SD) (Figure 2d). This suggests that control of the hemispindle branch length and number is altered in MCMBP-deficient parasites.

Mitotic spindles from parasites cultured either in the presence or absence of Shld1 were also imaged and measured (Figure 2e) (Appendix A). In the presence of Shld1, mitotic spindles form in an orderly fashion with branches that extend from each of the MTOC towards the opposing MTOC, and meet near the middle. By contrast, in the absence of Shld1, branches from the mitotic spindle appeared more heterogeneous in length, but do not appear to extend towards the other MTOC or meet near the middle of the two MTOCs. Moreover, mitotic spindles were 23% larger in the absence of Shld1 (691 nm ± 202 nm SD) than in the presence of Shld1 (529 nm ± 76 nm SD) (Figure 2f). This suggests that MCMBP-deficient parasites form larger mitotic spindles, where the organization and positioning of spindle branches is aberrant.

### 3.3. BODIPY TR Ceramide Stains the Nuclear Envelope of P. falciparum Imaged by U-ExM

*Plasmodium* undergoes mitosis without a breakdown of the nuclear envelope, and in doing so, the nuclear envelope provides a critical barrier for the compartmentalization of the nucleus from the cytoplasm. Therefore, the nuclear envelope integrity and remodeling are critical during *Plasmodium* mitosis. Despite the importance of nuclear envelope dynamics during schizogony and *Plasmodium* mitosis, there is currently no reliable marker of the *P. falciparum* nuclear envelope for microscopic visualization. Previous studies have localized a few nucleoporins (nups) to the nuclear envelope of either *P. falciparum* [16,39] or *P. berghei* [40], but their distribution and number is dynamic across the lifecycle, limiting the robustness of nups as nuclear envelope markers. Therefore, we wanted to identify a uniform, U-ExM compatible stain for the *P. falciparum* nuclear envelope to allow us to study nuclear envelope changes in the context of MCMBP-deficient parasites.

BODIPY TR ceramide (BODIPY TRc) is a commonly used fluorescent lipid stain that has previously been used to stain live parasites from multiple different parasite lifecycle stages, across *P. falciparum* and *P. berghei*, and imaged in both fixed and live-cell microscopy [40,41,42,43,44]. Despite its extensive use, BODIPY TRc has not previously been reported to stain the nuclear envelope of *P. falciparum*. We coupled BODIPY TRc with U-ExM, with BODIPY TRc staining occurring post-expansion. Remarkably, we found that the *P. falciparum* nuclear envelope is consistently and reliably labelled by BODIPY TRc (Figure 3a). In addition to staining the nuclear envelope, BODIPY TRc enabled observation of the RBC membrane, PVM, PPM, and endoplasmic reticulum as previously demonstrated when staining live parasites [43,44,45,46,47] (Appendix A). Together, we demonstrate that BODIPY TRc is the first *Plasmodium* nuclear envelope stain that is enabled by U-ExM.

To assess the relationship between intranuclear microtubule structures and the nuclear envelope, BODIPY TRc was coupled with U-ExM and tubulin staining (Figure 3a). The nuclei possessing each of the three microtubule structures display differently shaped nuclear envelopes. The nuclei with hemispindles show largely spherical nuclear envelopes, with some notable protrusions of the nuclear envelope to accommodate a hemispindle branch (Figure 3a). The nuclei with mitotic spindles display a marked pinching of the nuclear envelope around the site of the two MTOCs (Figure 3a). The nuclei with interpolar spindles show nuclear envelopes that look characteristically similar to the ‘dumbbell-shape’ of segregating nuclei in fission yeast (Figure 3a) [24]. Notably, the long and thin bridge region lacks DNA staining, and the interpolar spindles themselves are often present extremely close to the nuclear envelope (Figure 3a,b).

### 3.4. MCMBP-Deficient Parasites Form Anaphase Chromatin Bridges, Leading to Uneven DNA Segregation and Aneuploidy but Still Form Subpellicular Microtubules

It had previously been observed that MCMBP-deficient parasites form complex aberrant spindles, and it was hypothesized that chromatin connected multiple nuclei [25]. In the absence of a nuclear envelope marker, and at the resolution of conventional light microscopy, it could not be determined if these shared a single intact nuclear envelope. By visualizing interpolar spindles of MCMBP-deficient parasites at a higher resolution, we were able to update this model. Most prominently, in all interpolar spindles imaged after expansion, there was significant DNA staining inside the bridge region of nuclei connected by interpolar spindles (Figure 4) (Appendix A); this was reminiscent of chromatin bridges that occur during a defective anaphase of other organisms [48,49]. This contrasts with MCMBP^HADD^ parasites grown in the presence of Shld1, where DNA staining was not observed inside this bridge region (Figure 1b and Figure 3a). We sometimes observed dividing nuclei with interpolar spindles where each nucleus was of vastly different size, potentially indicating uneven DNA segregation in some nuclei following MCMBP knockdown (Appendix A). Moreover, we also observed interpolar spindles connecting MTOCs in nuclei that did not appear to be separating from each other at all (Appendix A). Collectively, this suggests that MCMBP-deficient parasites can form interpolar spindles, but are unable to evenly segregate DNA into daughter nuclei.

In MCMBP^HADD^ parasites grown in the absence of Shld1, we also frequently observed a single nuclear envelope that contained multiple microtubule structures and multiple MTOCs not connected by an interpolar spindle (Appendix A). Nuclei were observed that contained two hemispindles and two MTOCs, which likely represent nuclei where nuclear fission either did not occur or had occurred aberrantly (Figure 4) (Appendix A). Additionally, nuclei that contained two mitotic spindles and four MTOCs were also observed (Figure 4). This suggests that in the aberrant nuclei that contain two MTOCs after mitosis, both MTOCs can duplicate and form mitotic spindles in the same nucleus. Collectively, these observations suggest that MCMBP deficient parasites undergo uneven DNA segregation, leading to aneuploidy and nuclear fission defects, but that these defects do not inhibit further rounds of mitosis as shown previously [25].

Following multiple rounds of mitosis, *P. falciparum* commits to segmentation whereby nuclei and other organelles are enclosed into an individual PPM to form merozoites. In merozoites from segmented schizonts, the MTOC is no longer visible by NHS ester staining, and there are no visible intranuclear microtubule structures (Figure 1b). During segmentation, the only visible microtubules are the subpellicular microtubules, which extend from apical polar ring 2 at the apical end of the merozoite, to the basal complex as segmentation progresses (Figure 1b) [12,37,38,50]. MCMBP is not expressed while parasites are undergoing segmentation [25], but considering that MCMBP-deficient parasites show intranuclear microtubule defects (Figure 2 and Figure 4, Appendix A), it could be hypothesized that MCMBP knockdown causes downstream subpellicular microtubule defects. To determine whether MCMBP knockdown caused global microtubule defects, or only defects in the intranuclear microtubules, we analyzed MCMBP^HADD^ parasites grown in the presence of absent Shld1 that had been arrested post-segmentation with the schizont egress inhibitor E64 [35] by U-ExM.

In merozoites from MCMBP^HADD^, schizonts grown in the absence of Shld1 subpellicular microtubules were observed (Appendix A), suggesting that MCMBP-deficient parasites do not display a global microtubule polymerization defect. However, through NHS ester and BODIPY TRc staining, it was observed that some merozoites contained multiple sets of subpellicular microtubules (Appendix A). Additionally, merozoites had vastly differently sized nuclei and contained different varying numbers of rhoptries (Appendix A). This confirmed previous observations that merozoites from segmented MCMBP-deficient schizonts displayed aneuploidy [25]. Additionally, zoid merozoites, which lack DNA, were observed, and they too contained subpellicular microtubules (Appendix A).

Collectively, these results show that MCMBP-deficient parasites form anaphase chromatin bridges and fail to undergo correct nuclear fission, resulting in aneuploidy and the presence of multiple microtubule structures inside the same nucleus (Figure 5). Despite these severe nuclear defects, these parasites undergo further rounds of mitosis and still undergo segmentation and form subpellicular microtubules; this suggests that MCMBP knockdown does not cause global microtubule polymerization defects (Figure 5).

## 4. Discussion

The development of U-ExM and its application to *P. falciparum* parasites have allowed us to understand the functions of proteins and processes of *P. falciparum* to a level of detail not previously possible. We applied U-ExM in the context of MCMBP-deficient parasites to significantly refine our understanding of the function of this protein during the blood-stage replication of *P. falciparum.*

The U-ExM protocol used in this study is largely similar to previously published protocols [16,22,36], with the notable modification of changing the protein crosslinking (FA/AA) incubation step from 5 h to overnight, which significantly shortened Day 1 of the U-ExM protocol. In this study, we harvested parasites at multiple timepoints throughout the lifecycle, and all −Shld1 cultures were harvested 3 h after their +Shld1 counterparts due to the documented growth delay [25]. This shortening of Day 1 of the U-ExM protocol made the protocol far more practical, enabling the progressive study of different lifecycle stages in the same U-ExM experiment, rather than having to harvest different lifecycle stages as independent experiments.

In addition to highlighting new biology, our application of U-ExM allowed us to identify some potential drawbacks of this technique. Notably absent in all images of U-ExM parasites were the food vacuole and hemozoin crystal. Through the NHS ester staining of unexpanded parasites (Figure 1a), the likely location of the hemozoin crystal and food vacuole could be inferred due to a distinct lack of staining. By contrast, there was no indication on U-ExM parasites of where the hemozoin crystal or food vacuole membrane would be located based on NHS ester or BODIPY TRc staining. We hypothesize that the hemozoin crystal either does not get anchored, or does not expand with the gel, potentially limiting the utility of U-ExM for studies of hemoglobin catabolism and hemozoin biomineralization.

In this study, a wide range of nucleic acid stains were used on U-ExM parasites: DAPI, DRAQ5, SYBR Green, and SYTOX red. Notably, all these stains showed considerably more photobleaching than we would observe in unexpanded parasites; this was particularly notable for SYBR Green, which began visibly photobleaching almost immediately. The reason for this is not clear, but it should be noted that while in unexpanded MCMBP-deficient parasites a clear nuclear staining defect was observed [25], the same could not be readily observed in U-ExM parasites. At the concentrations used in this study, we found SYTOX red to be the brightest and most photostable of the nucleic acid stains used. Potentially related to the changes in chromatin was our inability to localize MCMBP by U-ExM. MCMBP has previously been localized in unexpanded parasites, showing nuclear and cytoplasmic foci [25]. Despite this, our attempts to localize MCMBP by U-ExM showed no significant signal. It has been noted previously that some antibodies appear to be incompatible with U-ExM [36], although the reasons for this are unclear, but this does not appear to be the case as the anti-HA antibody we used to detect MCMBP has previously been used successfully on U-ExM samples [16]. Given that the canonical role of MCMBP is to bind DNA, and we observe significant differences in the appearance of DNA in U-ExM parasites, it is possible that some DNA-binding proteins are not retained after U-ExM.

BODIPY TRc stained parasites presented in this study were stained post-expansion. We attempted to stain live cells with BODIPY TRc or include BODIPY TRc with the primary or secondary antibody incubations, but this uniformly resulted in extremely faint staining (data not shown). Additionally, we tried to stain parasites with Nile red, which has previously been shown to stain some organelles in unexpanded *P. falciparum* blood-stage parasites [45]; this was also unsuccessful (data now shown). Overall, this suggests that potentially large differences exist in the fluorescent stains that are compatible with unexpanded *P. falciparum* compared to with U-ExM.

The BODIPY TRc staining of schizonts allowed the visualization of the PPM of each merozoite, the PVM, RBC membrane, and nuclear envelope, but did not reveal any structures that could be characteristically identified as the apicoplast or mitochondrion, as have been identified by EM studies [12]. Additionally, both the surface and lumen of the rhoptries of merozoites stained very strongly with BODIPY TRc, which would support previous observations that the rhoptries contain membranous whorls [37,51]. To date, these membranous whorls have not been observed by light microscopy [52], highlighting the use of U-ExM coupled with BODIPY TRc for studying merozoite physiology.

We show that MCMBP knockdown results in the aberrant formation of all intranuclear microtubule structures, but not of subpellicular microtubules, which are formed when MCMBP is no longer expressed [25]. Moreover, the combination of NHS ester and BODIPY TRc with U-ExM allowed us to assess the MTOC and nuclear envelope at a level of detail reminiscent of electron microscopy. This also confirmed that in the blood-stage of *P. falciparum*, the MTOC spans the nuclear envelope, as previously reported [16]. The combination of BODIPY TRc and NHS ester staining allowed us to show that aneuploidy in MCMBP-deficient parasites is due to the formation of anaphase chromatin bridges and/or a lack of nuclear fission (Figure 5). We hypothesize that these lead to the downstream phenotypes we see of wildly varied nuclear size and zoid merozoites following cytokinesis.

The observation of anaphase chromatin bridges in MCMBP-deficient parasites is supported by the canonical function in the MCM complex [26]. In other organisms, the presence of MCMBP has been shown to promote dissociation between the MCM complex and chromatin, allowing the separation of sister chromatids [53,54,55]. Moreover, MCMBP of *P. falciparum* has been shown to interact with the members of the condensin complex structural maintenance of chromosomes (SMC) 2 and 4 [25]. SMC2 and SMC4 have canonical roles in chromosome condensation [56,57], and have recently been shown to be involved in *Plasmodium* chromosome separation [58]. The inhibition of SMC2 [59], SMC4 [60], and MCM complex member MCM7 [61] have all been shown to lead to the formation of anaphase chromatin bridges in other organisms. Therefore, we suggest that the observation of anaphase chromatin bridges in MCMBP-deficient *P. falciparum* may be caused by either an ability to properly separate sister chromatids, or a defect in the detachment of microtubules from chromosomes. Currently, the relationship between the formation of anaphase chromatin bridges and nuclear fission is unclear. However, given that the inhibition of SMC2, SMC4, and MCM7 causes anaphase chromatin bridges in organisms that undergo open mitosis, and therefore do not undergo nuclear fission, defective nuclear fission is not a pre-requisite for the formation of anaphase chromatin bridges.

While we observe anaphase chromatin bridges and multiple microtubule structures in a single nucleus following the depletion of MCMBP, using MCMBP^HADD^ parasites, these events do not occur in every round of mitosis. If this were the case, we would expect to see all DNA staining contained within a single, giant, nuclear envelope, but we do not. Given that the knockdown system used leads to the imperfect and uneven depletion of MCMBP, it is possible that the phenotypic heterogeneity we observe is a product of differing levels of MCMBP. MCMBP is likely essential for growth in the blood-stage of *Plasmodium* [62,63], and we hypothesize that the complete removal of MCMBP would lead to the formation of anaphase chromatin bridges and inhibited nuclear fission in every round of mitosis.

Our observations of microtubules, following U-ExM, were largely concordant with recent studies that also made measurements of branch and spindle lengths [16,22]. Measurements for mitotic spindle length, hemispindle branch length, and hemispindle branch number all reported similar results [16,22]. However, neither data set controlled for the number of nuclei per cell, and so it is currently unclear whether any of these measurements change later in the parasite lifecycle. However, one difference observed in our study was the presence of hemispindle-like branches in the nuclei connected by interpolar spindles (referred to as anaphase spindles in that study) [16]. Previous images have only observed the long interpolar branches connecting the MTOCs, without smaller branches in each nucleus [16]. Critically, a previous hypothesis suggested that the hemispindle formed as a remnant of the retraction of the interpolar spindle [64]. Our observation that the two seem to co-exist would suggest that this is not the case. Moreover, this suggests that whatever the function(s) of hemispindles are, they likely begin immediately following nuclear segregation and before nuclear fission.

Overall, this study provides insight into the poorly understood, yet therapeutically attractive and biologically fascinating, process of mitosis in *P. falciparum.* Our findings significantly further our understanding of the phenotype of parasites following the knockdown of MCMBP. Importantly, these insights were only possible because of the application of U-ExM to *P. falciparum*. Moreover, we developed BODIPY TRc as the first U-ExM-compatible stain to visualize the nuclear envelope, and used this to develop our understanding of both MCMBP-deficient parasites, and parasite physiology more broadly.

## Figures and Tables

**Figure 1 microorganisms-09-02306-f001:**
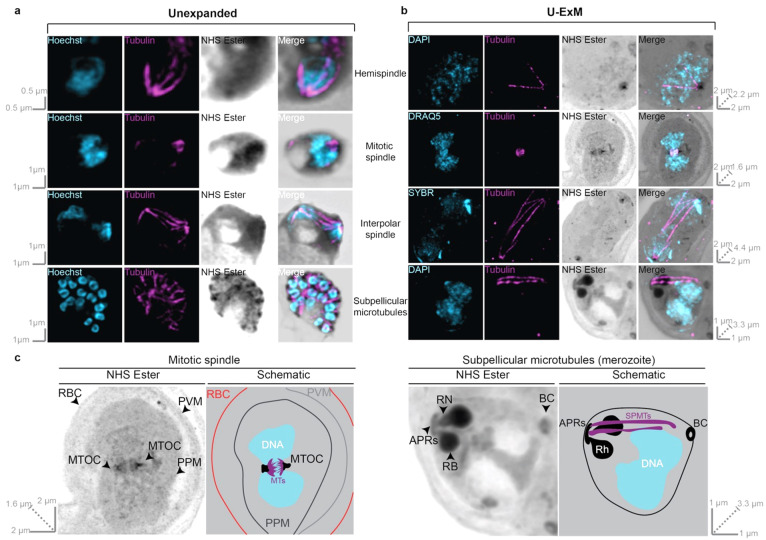
Comparison between microtubule structures visualized in unexpanded and U-ExM *P. falciparum* asexual blood-stage parasites. MCMBP^HADD^ parasites, cultured in the presence of Shld1, were imaged using super-resolution Airyscan microscopy after being prepared for regular immunofluorescence assay (**a**), or U-ExM (**b**). All parasites were stained with a nuclear stain (Hoechst, DAPI, DRAQ5, or SYBR in cyan), anti-tubulin (in magenta) and a protein stain (N-hydroxysuccinimide (NHS) Ester in greyscale). All previously identified blood-stage microtubule structures (hemispindle, mitotic spindle, interpolar spindle and subpellicular microtubules) were observed by both IFA and U-ExM. Images in (**a**) represent a single z-slice from a z-stack image, while images in (**b**) are maximum-intensity projections. Slice-by-slice videos of images in 1b found in Supplementary Videos 1-4. Scale bars as labelled in each image, solid bars = XY scale, dashed bar = combined depth of slices used for Z-projection. (**c**) Expanded and annotated view of NHS Ester channel from mitotic spindle and subpellicular microtubule images from (**b**) along with schematic interpretation of these images. Arrowheads point to NHS staining of interest. Colors in schematic: black = dense NHS Ester staining, grey = light NHS Ester staining, blue = DNA, purple = microtubules. RBC = Red blood cell membrane, PVM = Parasitophorous vacuole membrane, PPM = Parasite plasma membrane, MTOC = Microtubule organizing center, MTs = microtubules, APRs = Apical polar rings, RN = Rhoptry neck, RB = Rhoptry bulb, BC = Basal complex, Rh = rhoptry, SPMTs = Subpellicular microtubules.

**Figure 2 microorganisms-09-02306-f002:**
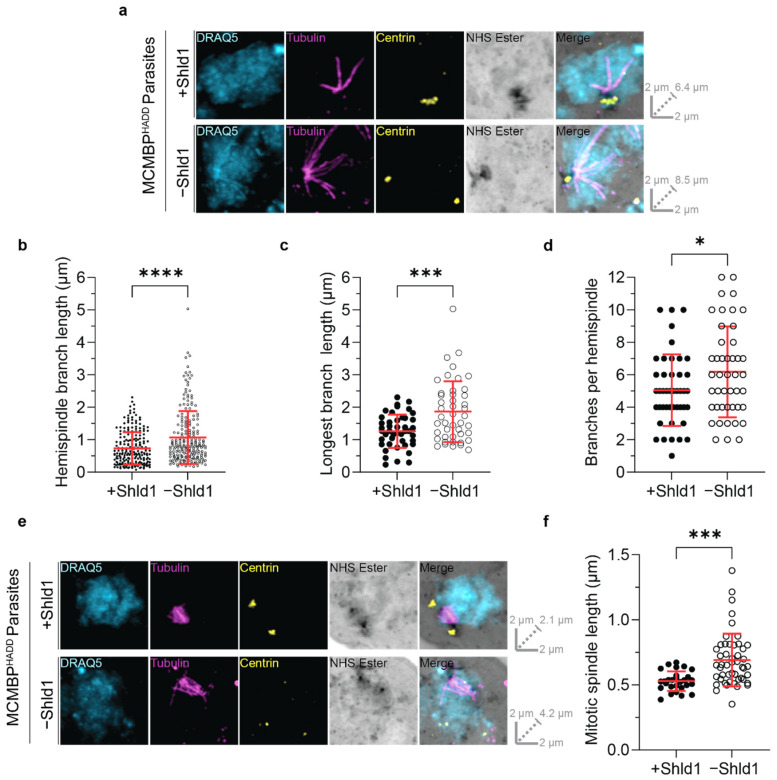
MCMBP-deficient parasites show defects in both mitotic spindle and hemispindle formation. MCMBP^HADD^ parasites were cultured [+]/[−] Shld1. Parasites were then prepared for U-ExM, stained with a nuclear stain (DRAQ5, in cyan), anti-tubulin (in magenta), anti-centrin (in yellow), and a protein stain (NHS Ester, in grayscale), and visualized using Airyscan microscopy. (**a**) Hemispindles were imaged, and the length of all hemispindle branches (**b**), of the longest branch in each individual hemispindle (**c**), and the total number of branches per hemispindle (**d**) were all measured. *n* = 221 hemispindle branches and 45 hemispindles for +Shld1, 214 hemispindle branches and 45 hemispindles for -Shld1 were measured across 3 biological replicates. (**e**) Mitotic spindles were imaged and their length (**f**), from one MTOC to another, was measured. *n* = 28 for +Shld1 and 49 for −Shld1, across 3 biological replicates. All distance measurements presented here have been estimated based on the average expansion factor of gels used in this study. Raw values can be found in Appendix A. (* *p* < 0.05, *** *p* <0.001, **** *p* < 0.0001 by unpaired two-tailed *t*-test, error bars = SD). All images are maximum intensity projections. Slice-by-slice videos of images are found in Videos S5–S8. Scale bars as labelled in each image, solid bars = XY scale, dashed bar = combined depth of slices used for Z-projection.

**Figure 3 microorganisms-09-02306-f003:**
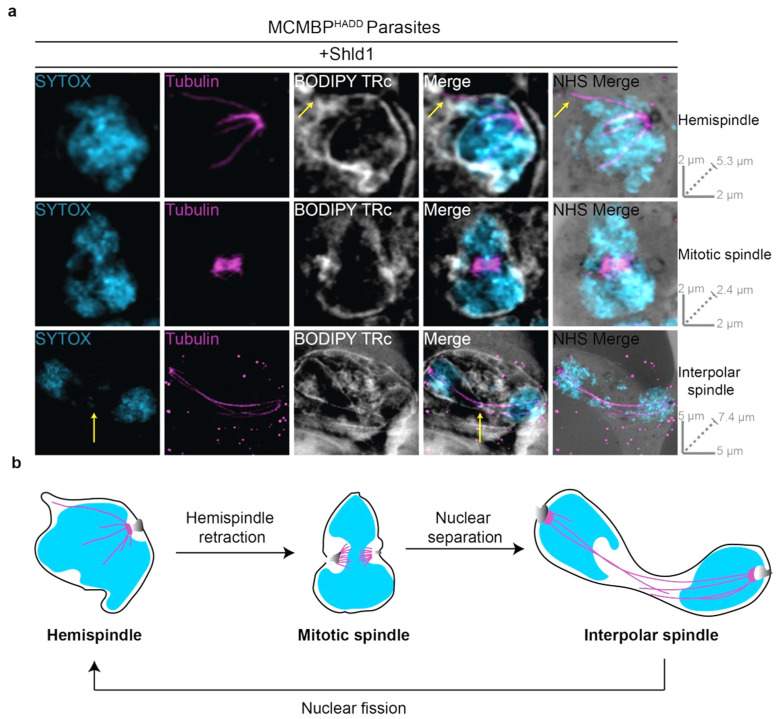
Nuclear envelope visualized using BODIPY TRc and U-ExM during mitosis of *P. falciparum* blood-stage. (**a**) MCMBP^HADD^ parasites were cultured in the presence of Shld1. Parasites were then prepared for U-ExM, stained with a nuclear stain (SYTOX, in cyan), anti-tubulin (in magenta), a membrane stain (BODIPY Texas Red ceramide (TRc), in white), and a protein stain (NHS Ester, in grayscale), and visualized using Airyscan microscopy. Hemispindle arrow indicates microtubule not associated with chromatin. Interpolar spindle arrow indicates chromatin-free bridge region. Images containing BODIPY TRc are average intensity projections, while those with NHS ester are maximum intensity projections. Slice-by-slice videos of images in 3a found in Videos S9–S11. Scale bars as labelled in each image, solid bars = XY scale, dashed bar = combined depth of slices used for Z-projection. (**b**) Model for the progression between observed microtubule structures as inferred from [16]. Hemispindles are first observed, but retract before formation of the mitotic spindle once DNA replication has occurred. After the formation of the mitotic spindle, two masses of DNA separate from each other but remain in a shared nuclear envelope with their MTOCs connected by the interpolar spindle. The nucleus then undergoes nuclear fission, separating the two separated DNA masses into daughter nuclei. Following nuclear fission, the hemispindle reforms and further rounds of mitosis occur.

**Figure 4 microorganisms-09-02306-f004:**
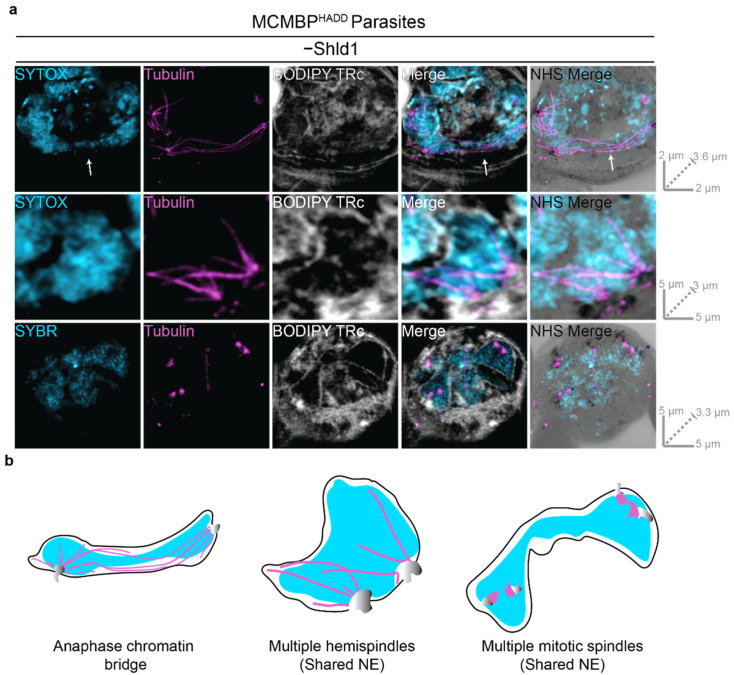
MCMBP-deficient parasites show defective interpolar spindles, uneven DNA segregation, and aneuploidy without cell cycle arrest. (**a**) MCMBP^HADD^ parasites were cultured in the absence of Shld1. Parasites were then prepared for U-ExM, stained with a nuclear stain (SYTOX or SYBR, in cyan), anti-tubulin (in magenta), a membrane stain (BODIPY TRc, in white), and a protein stain (NHS Ester, in grayscale), and visualized using Airyscan microscopy. Arrow indicates DNA staining in bridge-region. Images containing BODIPY TRc are average intensity projections, while those with NHS ester are maximum intensity projections. Slice-by-slice videos of images in 4a are found in Videos S12–S14. Scale bars as labelled in each image, solid bars = XY scale, dashed bar = combined depth of slices used for Z-projection. (**b**) Schematic representation of phenotypes overserved in Figure 4a.

**Figure 5 microorganisms-09-02306-f005:**
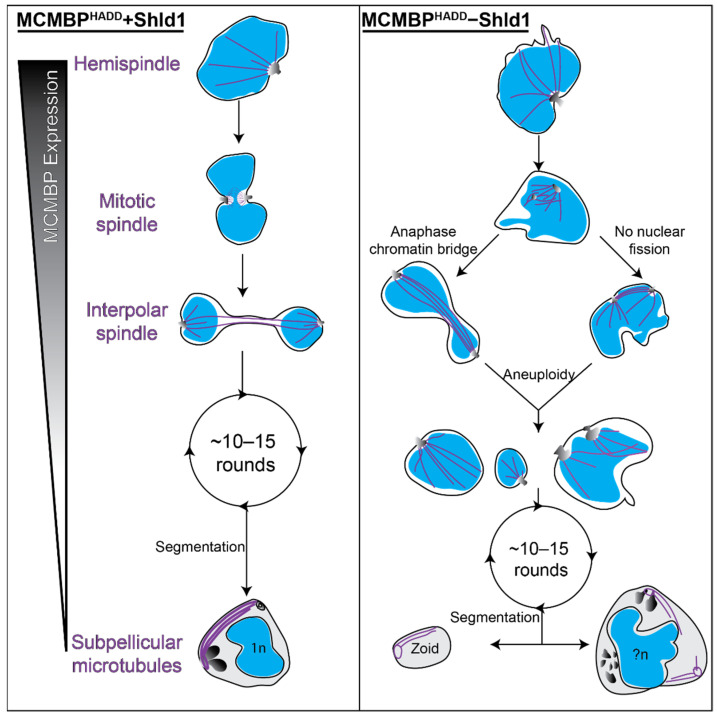
Aneuploidy in MCMBP-deficient parasites is likely caused by the formation of anaphase chromatin bridges and aberrant nuclear fission. Hypothetical model for the progression of mitosis and segmentation in MCMBP^HADD^ parasites either in the presence (left) or absence (right) of Shld1. In the presence of Shld1, nuclei with a single MTOC form a hemispindle. This hemispindle retracts and the MTOC duplicates and migrates to the opposing side of the nucleus to form the mitotic spindle. The two MTOCs then move away from each other, but remain connected by the interpolar spindle. The nucleus then undergoes nuclear fission in the DNA-free bridge region to form two daughter nuclei that each reform a hemispindle. The parasite undergoes multiple rounds of mitosis in this manner. MCMBP is no longer expressed when the parasite commits to segmentation and the formation of merozoites, where we observe subpellicular microtubules connecting the apical polar ring and basal complex. In the absence of Shld1, we observe aberrant hemispindles and mitotic spindles. After forming the mitotic spindle, these nuclei form interpolar spindles, but either form anaphase chromatin bridges and/or fail to undergo nuclear fission, which both lead to aneuploidy. Aneuploid nuclei continue to undergo further rounds of mitosis and do undergo segmentation. Segmentation, however, leads to the formation of cells of various sizes, including zoid parasites, which lack nuclei, and large merozoites that contain multiple sets of organelles and more than 1n DNA content. Blue = nuclei, purple = microtubules, greyscale = MTOC or rhoptries (in merozoites). n = number of genome copies, ?n = unknown/uneven genome content.

## Data Availability

Data that support the findings of this study are available from the corresponding author upon request. All graphs presented in this study display all individual data points. Staining of all representative images used in this study can be found in Appendix A.

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
