# Peer review of "Expansion Microscopy Reveals Plasmodium falciparum Blood-Stage Parasites Undergo Anaphase with A Chromatin Bridge in the Absence of Mini-Chromosome Maintenance Complex Binding Protein"

_microorganisms, 2021, doi:10.3390/microorganisms9112306_

Round 1
Reviewer 1 Report
Liffner and Absalon use U-ExM to analyze the phenotype of the conditional depletion of the MCM binding protein of the malaria-causing parasite Plasmodium falciparum. They use a membrane stain to visualize the nuclear envelop, which appears to yield satisfactory results in combination with U-ExM. While the images are clearly state-of-the-art, it is difficult to understand why the depletion of a protein that is most likely involved in genome duplication and/or maintenance should have an effect on microtubule organization and, hence, some of the quantifications are not immediately clear.
Major issues
This manuscript describes the phenotype of a MCM binding protein. But most experiments, quantifications, and text concern microtubule structure. Given that MCM is a critical helicase, it is unclear why the manuscript focusses on microtubules. The manuscript would strongly benefit from clearly stating throughout the text that the microtubule defects are likely secondary. Only in line 544 and later in line 583 this is discussed and the possible molecular connection is described.
Many claims of the manuscript are based on sometimes individual images. In absence of additional images or quantitative data, the manuscript will clearly benefit from a more balanced interpretation of the provided data.
Minor issues
Line 9, please revise first sentence of the abstract.
Throughout the text, starting in line 15, perhaps better use the term interpolar microtubules (or anaphase spindles as the title also speaks about anaphase).
Line 29, this sentence would benefit from revision.
Line 49, please provide references for this statement.
Line 55, please provide references for the different terms. Also, if there is already inconsistency, why introducing yet another term?
Line 261, this sentence appears to contradict the following sentences and a more conservative statement is appropriate.
Line 267-297, many of the findings described in the text corresponding to Fig. 1 are not easy to identify in the figure, especially in 1b where only one image per microtubule structure is shown. More images and/or additional labels like arrow heads would certainly help the reader.
Line 269, please highlight these membranous structures (RBC, PVM, PPM) to guide the reads that has no experience with U-ExM.
Line 268, this appears to be a strong statement for a very new technology. Many readers are certainly not familiar. And unlike EM, we do not have lots of data to compare these images.
Fig. 1b, by NHS stain it looks like there are two DAPI positive compartments. So the statement that NHS clearly identifies structures (line 261) is clearly misleading.
Line 301, the term ‘Airyscan microscopy’ implies that this is a certain type of microscopy, like Darkfield or light sheet microscopy. The correct term seems to be a microscope with an Airyscan detector and this should be changed throughout the text.
Line 325, this statement is based on a single picture and should be revised.
Line 378, in absence of a validated co-stain for the nuclear envelop we unfortunately do not know and this statement could be more balanced.
Line 387, please revise this sentence.
Line 411, DNA staining rather than chromatin staining.
Line 412, which part of the model was refined and how did the staining of the membranes help?
From the data shown in Fig. 3a, it is difficult to deduce any temporal information, which would inform on the progression.
Line 417, the diameter / area / volume of nuclei in these mutants was not quantified and this statement appears quite strong in absence of quantitative data.
Line 448, merozoites are not segmented, it is the schizont.
Line 452, if MCMBP is not expressed in these stages, why doing this experiment? Especially, since the effect on MTs may be a secondary as described further below (line 544).
Line 462, please revise this sentence and add ‘presence’.
Line 479, please reference a figure here.
Line 519, please explain ‘without needing to perform independent experiments`
Author Response
Response to Reviewer’s Comments
We thank the Reviewers for their comments and feedback. We provide a detailed and itemised description of the changes made to the manuscript to address the feedback. All changes can be cross-referenced following the line numbers listed in this document. Additionally, we used track changes in both the manuscript document and Supplementary Information so all areas where changes were made can be easily identified.
Reviewer # 1 major comments:
- This manuscript describes the phenotype of a MCM binding protein. But most experiments, quantifications, and text concern microtubule structure. Given that MCM is a critical helicase, it is unclear why the manuscript focusses on microtubules. The manuscript would strongly benefit from clearly stating throughout the text that the microtubule defects are likely secondary. Only in line 544 and later in line 583 this is discussed and the possible molecular connection is described.
We thank reviewer #1 for their comments and suggestions that helped us clarifying how MCMBP depletion could have an impact on microtubules organization during mitosis.
“it is unclear why the manuscript focusses on microtubules”
In the first study on the impact of MCMBP depletion on microtubule organization of P. falciparum parasites (Absalon and Dvorin, 2021) the authors discussed the possibility of DNA replication no longer coordinated with nuclear division resulting in the emergence of interconnected and extended spindle microtubules. In our study, we improved the visualization of microtubule using expansion microscopy allowing us to characterize the spatial organization of intranuclear microtubules in MCMBP deficient parasites (Fig 1 and 2). Furthermore, we utilized our newly developed nuclear envelope staining to confirm the absence of nuclear fission upon a second-round mitosis resulting in the presence of multiple MTOCs and mitotic spindles in a single nucleus (Fig 4).
Absalon, S.; Dvorin, J.D. Depletion of the mini-chromosome maintenance complex binding protein allows the progression of cytokinesis despite abnormal karyokinesis during the asexual development of Plasmodium falciparum. Cellular Microbiology. 2020, 23(3), e13284. doi: 10.1111/cmi.13284.
“The manuscript would strongly benefit from clearly stating throughout the text that the microtubule defects are likely secondary.”
Reviewer 1 raised an excellent point that we insufficiently highlighted in the first version of our manuscript which is why Plasmodium parasite is such a powerful model to study MCMBP function. Unlike other eukaryotic organism in which MCMBP has been functionally studied, P. falciparum parasites lack cell cycle checkpoints, such as the spindle assembly checkpoint which blocks mitotic exit and the next round of DNA replication upon spindle disruption. Therefore, we have the unique ability to monitor MCMBP depletion defects throughout the entire process of mitosis, notably the impact of MCMBP depletion on intranuclear microtubules organization. Moreover, it has been shown that another key component of the pre-replicative complex named CDC6 is required for MTOC assembly and function. Depletion of CDC6 in human cells resulted in an increase of microtubule formation (Lee et al 2007), PMID: 28827311). Therefore we added the following text to the introduction:
“…In a recent study, the authors generated a transgenic parasite line for inducible knockdown where the destabilization domain system was incorporated into MCMBP (MCMBPHADD) [25]. Using this system, the addition of a molecule called Shield-1 (Shld1) results in wildtype expression of MCMBP, while the absence of Shld1 results in specific MCMBP knockdown. MCMBP deficient parasites were able to undergo DNA replication, but showed severe microtubule defects and aneuploidy; attributed to defective DNA segregation [25]. Given then canonical role of MCMBP in DNA replication, it is likely that the microtubule defects observed are downstream, rather than direct, effects of MCMBP depletion…”
“Here, we use U-ExM to study spatial organization of intranuclear microtubules and nuclear division in the context of MCMBP deficient parasites. While the effects of MCMBP knockdown on microtubules are likely downstream of MCMBP function, Plasmodium continues DNA replication even the presence of these defects due to a lack of canonical cell cycle checkpoints [29]. This provides the unique ability to study the ongoing impact of both MCMBP knockdown, and microtubule dysregulation in parasite mitosis. Additionally, we develop the first U-ExM compatible nuclear envelope stain. We use U-ExM to show that MCMBP deficient parasites display defective hemispindles and mitotic spindles. Additionally, we couple U-ExM and nuclear envelope staining to show that MCMBP deficient parasites form anaphase chromatin bridges, which leads to aneuploidy and the presence of multiple microtubule structures contained within the same nuclear envelope.”
Lee, I.; Kim, G.S.; Bae, J.S.; Kim, J.; Rhee, K.; Hwang, D.S. The DNA replication protein Cdc6 inhibits the microtubule-organizing activity of the centrosome. Journal of Biological Chemistry. 2017 292(39):16267-16276. doi: 10.1074/jbc.M116.763680
“Only in line 544 and later in line 583 this is discussed and the possible molecular connection is described.”
MCMBP function in eukaryotic cells is yet to be determined and, in our discussion, we highlighted one favoured hypothesis that was first described in Nishiyama et al 2011 where they demonstrated that in human cell line MCMBP promotes dissociation between the MCM complex and chromatin in late S phase (line 562-563). In a future study, we will aim to decipher the molecular mechanism driving DNA replication and segregation and define the precise role of MCMBP in this mechanism. We will investigate the hypothesis discussed in line 571 that in absence of MCMBP, MCM complex remains onto the chromatin leading to aberrant chromatin architecture and microtubule organization, which is out of scope of our current manuscript.
Nishiyama, A.; Frappier, L.; Méchali, M. MCM-BP regulates unloading of the MCM2-7 helicase in late S phase. Genes & Development. 2011 25(2):165-75. doi: 10.1101/gad.614411.
- Many claims of the manuscript are based on sometimes individual images. In absence of additional images or quantitative data, the manuscript will clearly benefit from a more balanced interpretation of the provided data.
We thank reviewer #1 for their suggestion and we made several changes throughout the manuscript to ensure a balanced interpretation and justification of our data, please see our response to minor revisions line: 49-51,263-269, 270-272, 325, 378, 417 and 452. In addition, we extracted images from plotted data into supplementary figures 3 and 4 to provide additional images of both hemispindles and mitotic spindles that were used in our quantitative analysis.
Reviewer # 1 minor comments:
- Line 9, please revise first sentence of the abstract.
There appears to be a discrepancy between the line numbers observed by Reviewer # 1, and those that we observe. Due to this, in some instances it is not clear what line exactly Reviewer #1 is asking us to revise. We have done our best to infer, but apologise if we’ve missed the error they were trying to highlight. The error the beginning of the abstract has been corrected. The sentence now reads (lines 10 – 11):
“The malaria parasite Plasmodium falciparum undergoes closed mitosis, which occurs within an intact nuclear envelope, and differs significantly from its human host.”
- Throughout the text, starting in line 15, perhaps better use the term interpolar microtubules (or anaphase spindles as the title also speaks about anaphase).
In our opinion, referring to the microtubule structure as the interpolar spindle best reflects its defining feature of connecting two MTOCs at opposing ends of a dividing nucleus. Please see the response to minor comment # 5 for a more detailed explanation of this.
- Line 29, this sentence would benefit from revision.
This sentence has been revised and now reads (lines 23-24):
“Collectively, this study refines our understanding of MCMBP-deficient parasites and highlights the utility of U-ExM coupled with a nuclear envelope stain for studying mitosis in P. falciparum.”
- Line 49, please provide references for this statement.
A reference has been included in this sentence. Additionally, this sentence has been revised to reflect what previously was a slight overstatement of currently published results. Previously the sentence implied that the hemispindle retracts around the time of DNA replication. While this has been hypothesised, it isn’t currently supported by published experimental data and so the wording of this sentence has been changed as follows (lines 49-51):
“It has been shown that during the first mitosis, the hemispindle retracts, the MTOC duplicates, and the mitotic spindle is formed [16].”
Reference 16: Simon, C.S.; Funaya, C.; Bauer, J.; Voß, Y.; Machado, M.; Penning, A.; Klaschka, D.; Cyrklaff, M.; Kim, J.; Ganter, M.; et al. An extended DNA-free intranuclear compartment organizes centrosomal microtubules in Plasmodium falciparum. Life Science Alliance 2021, 4, e202101199, doi:10.26508/lsa.202101199.
- Line 55, please provide references for the different terms. Also, if there is already inconsistency, why introducing yet another term?
We agree with the reviewer that it is indeed confusing to use a new term, but in our opinion interpolar spindle is the most appropriate name for the microtubule structure. The structures that we call interpolar spindles have previously been called multiple names including astral microtubules, complete hemispindles, and extended spindles. We use the term interpolar spindle because in our opinion, this best reflects the defining feature of this microtubule structure, which is connecting two distant MTOCs. The term astral microtubules is misleading as they do not form from an aster. Complete hemispindle is inaccurate, as it is based on the incorrect assumption that it forms from the fusion of two hemispindles. Extended spindle does describe that this structure is typically longer than the branches of the hemispindle, but does not make clear that it connects two MTOCs. Moreover, interpolar spindle is a term that has been used previously to describe microtubule structures that connect two MTOCs but don’t attach to kinetochores (Mastronarde et al., 1993).
Mastronarde, D.N.; McDonald, K.L.; Ding, R.; McIntosh, J.R. Interpolar spindle microtubules in PTK cells. Journal of Cell Biology 1993, 123 (6), 1475-1489.
We have clarified in the text the other names previously ascribed to interpolar spindles, with corresponding references, and the sentence now reads as follows (Line 55-59):
“The nomenclature around this microtubule structure is inconsistent, having previously been named astral microtubules [14], complete hemispindles [15], and extended spindles [16], but in this study they will be referred to as interpolar spindles as their defining feature is connecting two distant MTOCs”
Reference 14: Gerald, N.; Mahajan, B.; Kumar, S. Mitosis in the Human Malaria Parasite Plasmodium falciparum. Eukaryotic Cell 2011, 10, 474-482, doi:doi:10.1128/EC.00314-10.
Reference 15: Read, M.; Sherwin, T.; Holloway, S.P.; Gull, K.; Hyde, J.E. Microtubular organization visualized by immunofluorescence microscopy during erythrocytic schizogony in Plasmodium falciparum and investigation of post-translational modifications of parasite tubulin. Parasitology 1993, 106 ( Pt 3), 223-232, doi:10.1017/s0031182000075041.
Reference 16: Simon, C.S.; Funaya, C.; Bauer, J.; Voß, Y.; Machado, M.; Penning, A.; Klaschka, D.; Cyrklaff, M.; Kim, J.; Ganter, M.; et al. An extended DNA-free intranuclear compartment organizes centrosomal microtubules in Plasmodium falciparum. Life Science Alliance 2021, 4, e202101199, doi:10.26508/lsa.202101199.
- Line 261, this sentence appears to contradict the following sentence and a more conservative statement is appropriate.
We agree, these two statements seem contradictory. The section has been worded to better reflect what can be observed NHS staining on unexpanded parasites as follows (Line 269-275):
“…In unexpanded parasites NHS ester staining did not produce a staining pattern that obviously represented a particular organelle. Despite no obvious demarcation of organelles, NHS ester staining appeared slightly denser in the chromatin-free region of the nucleus (Figure 1a), which has previously been shown to contain the microtubule organizing center (MTOC) [16]. Additionally, in segmented schizonts NHS ester staining appeared denser at the apical tip of merozoites, likely corresponding to the merozoite secretory organelles rhoptries, micronemes, or dense granules (Figure 1a).”
Reference 16: Simon, C.S.; Funaya, C.; Bauer, J.; Voß, Y.; Machado, M.; Penning, A.; Klaschka, D.; Cyrklaff, M.; Kim, J.; Ganter, M.; et al. An extended DNA-free intranuclear compartment organizes centrosomal microtubules in Plasmodium falciparum. Life Science Alliance 2021, 4, e202101199, doi:10.26508/lsa.202101199.
- Line 267-297, many of the findings described in the text corresponding to Fig. 1 are not easy to identify in the figure, especially in 1b where only one image per microtubule structure is shown. More images and/or additional labels like arrow heads would certainly help the reader.
This was an oversight on our behalf. Figure 1 has now been modified to include a panel 1c that contains an annotated expanded view of the NHS channel and schematics from the mitotic spindle and merozoite images from Figure 1 . These have been labelled to show the structures we can observe. The figure legend has been modified to reflect this inclusion as follows (Line 315 – 329):
“Figure 1: Comparison between microtubule structures visualized in unexpanded and U-ExM P. falciparum asexual blood-stage parasites.
MCMBPHADD parasites, cultured in the presence of Shld1, were imaged using super-resolution Airyscan microscopy after being prepared for regular immunofluorescence assay (a), or U-ExM (b). All parasites were stained with a nuclear stain (Hoechst, DAPI, DRAQ5, or SYBR in cyan), anti-tubulin (in magenta) and a protein stain (N-hydroxysuccinimide (NHS) Ester in greyscale). All previously identified blood-stage microtubule structures (hemispindle, mitotic spindle, interpolar spindle and subpellicular microtubules) were observed by both IFA and U-ExM. Images in (a) represent a single z-slice from a z-stack image, while images in (b) are maximum-intensity projections. Slice-by-slice videos of images in 1b found in Supplementary Videos 1-4. Scale bars as labelled in each image, solid bars = XY scale, dashed bar = combined depth of slices used for Z-projection. (c) Expanded and annotated view of NHS Ester channel from mitotic spindle and subpellicular microtubule images from (b) along with schematic interpretation of these images. Arrowheads point to NHS staining of interest. Colors in schematic: black = dense NHS Ester staining, grey = light NHS Ester staining, blue = DNA, purple = microtubules. RBC = Red blood cell membrane, PVM = Parasitophorous vacuole membrane, PPM = Parasite plasma membrane, MTOC = Microtubule organizing center, MTs = microtubules, APRs = Apical polar rings, RN = Rhoptry neck, RB = Rhoptry bulb, BC = Basal complex, Rh = rhoptry, SPMTs = Subpellicular microtubules.”
Additionally, we have included two new supplementary figures that contain further examples of both hemispindles (Supplementary Figure 3) and mitotic spindles (Supplementary Figure 4) in MCMBPHADD parasites both in the presence and absence of Shld1 whose lengths were quantified in Figure 2. The figure legends of these supplementary figures read as follows:
“Supplementary Figure 3. Examples of hemispindles in MCMBPHADD parasites in the presence and absence of Shld1.
Representative Airyscan microscopy images of hemispindles from MCMBPHADD parasites cultured either [+]/[-] Shld1 and prepared for U-ExM after staining with a nuclear stain (DRAQ5, in cyan), anti-tubulin (in magenta), anti-centrin (in yellow), and a protein stain (NHS Ester, in grayscale. All images are maximum intensity projections. Scale bars as labelled in each image, solid bars = XY scale, dashed bar = combined depth of slices used for Z-projection.”
“Supplementary Figure 4. Examples of mitotic spindles in MCMBPHADD parasites in the presence and absence of Shld1.
Representative Airyscan microscopy images of mitotic from MCMBPHADD parasites cultured either [+]/[-] Shld1 and prepared for U-ExM after staining with a nuclear stain (DRAQ5, in cyan), anti-tubulin (in magenta), anti-centrin (in yellow), and a protein stain (NHS Ester, in grayscale. All images are maximum intensity projections. Scale bars as labelled in each image, solid bars = XY scale, dashed bar = combined depth of slices used for Z-projection.”
- Line 269, please highlight these membranous structures (RBC, PVM, PPNM) to guide the reads that has no experience with U-ExM.
Please refer to comment 7. Where this concern was addressed.
- Line 268, this appears to be a strong statement for a very new technology. Many readers are certainly not familiar. And unlike EM, we do not have lots of data to compare these images.
This statement was intended to be a direct comparison between NHS ester staining in unexpanded, where we could not clearly identify any structures based on shape, and U-ExM parasites where we could now identify structures that were previously unclear. We did not mean to imply that it was immediately obvious what all the observed structures were. We have reworded the sentence to make this clearer as follows (Line 276-278):
“Despite its unclear staining in unexpanded parasites, U-ExM parasites stained with NHS ester allowed the identification of many intracellular structures that were not recognizable in unexpanded parasites”
- 1b, by NHS stain it looks like there are two DAPI positive compartments. So the statement that NHS clearly identifies structure (line 261) is clearly misleading.
While it may appear that there are two DAPI positive compartments by NHS ester, this is a product of the shape of the nucleus and the maximum intensity projection. Moreover, we did not state in the text that we can identify the nucleus by NHS staining. The nuclei of merozoites from fully segmented schizonts adopt roughly ‘kidney-bean’ shape, with a depression in the middle of the nucleus (Rudlaff et al., 2020). In the merozoite pictured in Figure 1b, it can be seen that the cytoplasm of the merozoite is more darkly stained with NHS ester than the region where the DAPI is, with the exception of a darker strip that the reviewer is interpreting as a separate compartment. What this darker segment actually represents is the depression in the merozoite nucleus, where rather than the less NHS dense nucleus, we are actually seeing the more NHS dense merozoite cytoplasm. In a maximum intensity projection, only the voxels with the brightest signal at that XY position will be represented, which are those from the cytoplasm because the NHS signal is more intense there. In Supplementary Video 4, this image is displayed in a ‘slice-by-slice view’ and it can be seen that these are not two separate compartments.
Rudlaff, R.M.; Kraemer, S.; Marshman, J.; Dvorin, J.D. Three-dimensional ultrastructure of Plasmodium falciparum throughout cytokinesis. PLoS Pathogens 2020, 16(6) e1008687.
- Line 301, the term ‘Airyscan microscopy’ implies that this is a certain type of microscopy, like Darkfield or light sheet microscopy. The correct term seems to be a microscope with an Airyscan detector and this should be changed throughout the text.
While confocal microscopy using an Airyscan detector would also be appropriate, Airyscan microscopy is a term that is already commonly used in studies using the technique in Plasmodium (Liffner & Frölich et al., 2020; Rudlaff et al., 2019; Connelly et al., 2021). Additionally, the term Airyscan microscopy is also used outside Plasmodium, with some studies containing it in their title (Korobchevskaya et al., 2017; Romero et al., 2020). While it is more a matter of semantics, we suggest that Airyscan microscopy is a certain kind of microscopy. Airyscan processing can only be performed if you acquire your images with a particular detector, a particular objective lens, and do so with particular settings, rather than an algorithm that can be applied to any image.
Liffner, B; Frölich, S; et al. PfCERLI1 is a conserved rhoptry associated protein essential for Plasmodium falciparum merozoite invasion of erythrocytes. Nature Communications 2020. 11, 1141.
Rudlaff, R.M.; Kraemer, S.; Streva, V.A.; Dvorin, J.D. An essential contractile ring protein controls cell division in Plasmodium falciparum. Nature Communications 2019. 10, 2181.
Connelly, S.V.; et al. Restructured Mitochondrial-Nuclear Interaction in Plasmodium falciparum Dormancy and Persister Survival after Artemisinin Exposure. mBio 2021. 12 (3).
Korobchevskaya, K.; Lagerholm, B.C.; Colin-York, H.; Fritzsche, M. Exploring the Potential of Airyscan Microscopy for Live Cell Imaging. Photonics 2017, 4, 41. https://doi.org/10.3390/photonics4030041
Romero, I.C.; Urban, M.A.; Punyasena, S.W. Airyscan Superresolution microscopy: A high-throughput alternative to electron microscopy for the visualiza
tion and analysis of fossil pollen. Review of paleobotany and palynology 2020. 276.
- Line 325, this statement is based on a single picture and should be revised.
Due to the aforementioned issue with line numbering, it is not clear whether the statement based on a single picture referred to centrin localizing to the cytoplasmic side of the MTOC, or MCMBP KD parasites having misplaced centrin foci and aberrant MTOCs. In either case, supplementary figures showing further examples of hemispindles (Supplementary Figure 3) and mitotic spindles (Supplementary Figure 4) in MCMBPHADD either in the presence of absence of Shld1 have been added, which contain further examples of both centrin localizing to the cytoplasmic side of the MTOC, and of aberrant MTOCs and centrin staining in MCMBP KD parasites.
- Line 378, in absence of a validated co-stain for the nuclear envelope we unfortunately do not know and this statement could be more balanced.
We respectfully disagree with the reviewer’s interpretation that it cannot be reliably stated that we are visualizing the nuclear envelope with BODIPY TRc. It is firmly established that DNA stains, such as SYTOX and DRAQ5 (which were used in experiments with BODIPY TRc) stain DNA in the nuclei of P. falciparum. It is also known that the nuclei are enclosed by a nuclear envelope. Therefore, the presence of a lipid-stain, in BODIPY TRc, at the periphery of the DNA staining can firmly and confidently be inferred to be the nuclear envelope. For different microscopy techniques, such as EM, the nuclear envelope has never been validated with a uniform co-stain in Plasmodium yet there are many published examples of Plasmodium EM that concern the nuclear envelope (Kehrer et al., 2018; Hanssen et al., 2013; Rudlaff et al., 2020; Weiner et al., 2011). Moreover, the ER forms the outer membrane of the nuclear envelope and in Supplementary Figure 5 (previously Supplementary Figure 3), we demonstrate that BODIPY TRc colocalizes with the portion of the ER marker BIP that encases the DNA stain. It is accurate to say that we cannot currently confirm whether BODIPY TRc stains in inner or outer membrane of the nuclear envelope, or both, but in our opinion, the evidence provided this study firmly establishes that BODIPY TRc stains, amongst other membranes, the nuclear envelope.
Rudlaff, R.M.; Kraemer, S.; Marshman, J.; Dvorin, J.D. Three-dimensional ultrastructure of Plasmodium falciparum throughout cytokinesis. PLOS Pathogens 2020, 16, e1008587, doi:10.1371/journal.ppat.1008587.
Kehrer, J.; Kuss, C.; Andres-Pons, A.; Reustle, A.; Dahan, N.; Devos, D.; Kudryashev, M.; Beck, M.; Mair, G.R.; Frischknecht, F. Nuclear Pore Complex Components in the Malaria Parasite Plasmodium berghei. Scientific Reports 2018, 8, 11249, doi:10.1038/s41598-018-29590-5.
Hanssen, E.; Dekiwadia, C.; Riglar, D.T.; Rug, M.; Lemgruber, L.; Cowman, A.F.; Cyrklaff, M.; Kudryashev, M.; Frischknecht, F.; Baum, J.; et al. Electron tomography of Plasmodium falciparum merozoites reveals core cellular events that underpin erythrocyte invasion. Cellular Microbiology 2013, 15, 1457-1472, doi:https://doi.org/10.1111/cmi.12132.
Weiner, A.; Dahan-Pasternak, N.; Shimoni, E.; Shinder, V.; von Huth, P.; Elbaum, M.; Dzikowski, R. 3D nuclear architecture receleas coupled cell cycle dynamics of chromatin and nuclear pores in the malaria parasite Plasmodium falciparum. Cellular Microbiology 2011, 13, 967-977, doi: https://doi.org/10.1111/j.1462-5822.2011.01592x
- Line 387, please revise this sentence.
The spelling/grammar errors in this sentence have been corrected.
- Line 411, DNA staining rather than chromatin staining.
This sentence, along with a handful of other instances where “chromatin staining” or “chromatin stain” were used have been changed to either say either “DNA staining/stain” or “nucleic acid staining/stain”
- Line 412, which part of the model was refine and how did the staining of the membranes help?
As mentioned in minor comment 17, the way some of this text was written conflated some of what had been shown temporally from other studies, with what was shown directly in this work. The way this sentence was written was misleading and so it has since been removed.
While our model for the progression of mictoubule structures in wildtype parasites did not change significantly in this study, our model for the phenotype in MCMBP deficient parasites did. In the previous study on MCMBP deficient parasites (Absalon & Dvorin, 2021) the authors hypothesized that: ”It is possible that in the absence of PfMCMBP, DNA replication is no longer coordinated with karyokinesis resulting in the emergence of interconnected and extended spindle microtubules”. In our current manuscript we utilized BODIPY-TR-ceramide stain (Figure 3) to demonstrate that the absence of nuclear division or karyokinesis upon the second round of DNA replication. In addition, we coupled immunostaining of microtubules with U-ExM (Figure 2), we demonstrated that not only MTOC were connected with interpolar microtubules but hemispindles and mitotic spindles were aberrant (figure 2). Altogether our quantitative analysis of MCMBP-deficient parasite using U-ExM refine the current model of MCMBP depletion during Plasmodium mitosis.
Absalon, S.; Dvorin, J.D. Depletion of the mini-chromosome maintenance complex binding protein allows the progression of cytokinesis despite abnormal karyokinesis during the asexual development of Plasmodium falciparum. Cellular Microbiology 2021, 23, e13284, doi:https://doi.org/10.1111/cmi.13284.
- From the data shown in Fig. 3a, it is difficult to deduce any temporal information, which would inform on the progression.
We agree with the reviewer that the data presented in this study does not provide any temporal information regarding the progression from hemispindle to mitotic spindle to interpolar spindle. It was a misrepresentation on our behalf to suggest that our data directly suggested this. We have included a reference into the legend into Figure 3 to clarify that this temporal transition has previously been determined using live-cell microscopy (Simon et al., 2021) and we are retrofitting our U-ExM images onto what has already been established temporally.
Simon, C.S.; Funaya, C.; Bauer, J.; Voß, Y.; Machado, M.; Penning, A.; Klaschka, D.; Cyrklaff, M.; Kim, J.; Ganter, M.; et al. An extended DNA-free intranuclear compartment organizes centrosomal microtubules in Plasmodium falciparum. Life Science Alliance 2021, 4, e202101199, doi:10.26508/lsa.202101199.
- Line 417, the diameter / area / volume of nuclei in these mutants was not quantified and this statement appears quite strong in absence of quantitative data.
We agree, the previous wording was likely an overstatement of the data presented in this study. We have altered the sentence as follows, to better reflect that this not observed in all nuclei and to soften the conclusions: “We sometimes observed dividing nuclei with interpolar spindles where each nucleus was of vastly different size, potentially indicating uneven DNA segregation in some nuclei following MCMBP knockdown (Supplementary Figure 6).”
- Line 448, merozoites are not segmented it is the schizont.
This has been changed to read: “In merozoites from segmented schizonts…”.
- Line 452, if MCMBP is not expressed in these stages, why doing this experiment? Especially, since the effect on MTs may be a secondary as described further below (line 554)
While this experiment likely tells us relatively little about the function of MCMBP itself, understanding the downstream effects of defects in DNA replication can help inform us more broadly about parasite physiology. Additionally, the phenotype in schizonts was studied in the previous study on MCMBP (Absalon & Dvorin, 2020), so this provided some continuity. But we do agree that the lack of MCMBP expression likely mean anything we observe is a downstream effect of MCMBP knockdown, which is why this work was restricted to the Supplementary Figures.
Absalon, S.; Dvorin, J.D. Depletion of the mini-chromosome maintenance complex binding protein allows the progression of cytokinesis despite abnormal karyokinesis during the asexual development of Plasmodium falciparum. Cellular Microbiology 2021, 23, e13284, doi:https://doi.org/10.1111/cmi.13284.
- Line 462, please revise this sentence and add ‘presence’
This sentence has been revised to “MCMBPHADD parasite were cultured in the absence of Shld1.” As this figure only contains images from parasites without Shld1.
- Line 479, please reference a figure here
A reference to Supplementary Figure 7 (previously Supplementary Figure 5) has been included.
- Line 519, please explain ‘without need to perform independent experiments’
We thank the reviewer for pointing this out, “without need to perform independent experiments” could be interpreted as our protocol not requiring biological replicates, and this was certainly not our intention. As explained below, this term was meant to relate to harvesting different timepoints as independent experiments, rather than multiple timepoints from the same parasite culture, because of the impracticality of the original U-ExM protocol.
The original U-ExM protocol, the first day of the protocol required ~13 hours in the lab if only harvesting at a single timepoint. Therefore, if you wanted to harvest at multiple timepoints, say early and late schizonts (separated by ~5 hours), you would have a minimum of ~18 hours in the lab, followed by another ~10 hour day the following day. The way the protocol has been modified for our study requires only ~1 hour on the first day, allowing much more practical harvest of the same parasite cultures at different stages of the lifecycle. Given the significant impracticality of such long days in the lab if harvesting at multiple timepoints, it would often be easier to perform these two timepoints as independent experiments.
We have reworded this sentence to clarify this:
“This shortening of day 1 of the U-ExM protocol made the protocol far more practical, enabling the progressive study of different lifecycle stages in the same U-ExM experiment, rather than having to harvest different lifecycle stages as independent experiments.”
Reviewer 2 Report
Over the past years, the size of the parasite nucleus has made studies of nuclear proteins difficult due to the lack of resolution of conventional microscopy. To overcome this roadblock the authors notably employed U-ExM - a technique recently adapted to Plasmodium parasites and cleverly used the BODIPY stain to visualize the nuclear envelope.
This manuscript is interesting, well written and documented. I commend the authors for their detailed methodology section - this study will surely become a methodological reference for U-ExM studies of nuclear proteins.
- While I understand that the authors have tested different conditions of nuclear stain it was not clear why they alternated between Hoechst, SYBR, DRAQ5 and SYTOX in the manuscript figures. Indeed, in fig 1b for instance, we see three of these dyes – one in each panel, without any explanation. Could the authors please clarify that?
- BODIPY TRc should be referred to as a stain rather than marker.
- Lines 449-452: (…) microtubules which extend from apical polar ring 2 … to the basal complex (Fig 1b.) --> This statement does not seem to be based on data shown in Fig 1b. Could the authors please rectify that?
- Some of the figures (for example sup fig 5) were hard to read - the authors should label structures and indicate clearly the structures that they refer to in the text (eg arrows, asterisks, …).
Line 312. Remove “due to”
Line 550. “data no shown”; not
Line 338 “parasites cultures”; cultured
Author Response
Reviewer # 2 comments:
- While I understand that the authors have tested different conditions of nuclear stain it was not clear why they alternated between Hoechst, SYBR, DRAQ5 and SYTOX in the manuscript figures. Indeed, in fig 1b for instance, we see three of these dyes – one in each panel, without any explanation. Could the authors please clarify that?
These differences in nuclear stain presented in this paper are for a number of different reasons. Firstly, we wanted to test a range of nuclear stains with different emission wavelengths to provide more adaptability with other stains. Here we have represented DNA stains with emission maxima of 461 nm (DAPI), 522 nm (SYBR Green I), 682 nm (SYTOX Deep Red) and 697 nm (DRAQ5). Secondly, we used different nuclear stains based on what other stains we were using in that experiment. Images where parasites were only stained with 3 markers (Tubulin, NHS Ester Atto 594, nuclear stain) were all stained with DAPI. However, DAPI and NHS Ester Atto 594 were not easily compatible when we wanted to do stain with 4 markers as we typically saw bleed through of the NHS Ester Atto 594 into the 647 nm channel. Therefore, all images that had four markers used either SYBR Green I, DRAQ5, or SYTOX Deep Red depending on what the corresponding stains and lifecycle stages were. For images where we stained with anti-tubulin (Alexa Fluor 594) and anti-centrin (Alexa Fluor 488), the only nuclear stain that didn’t show significant bleed through from tubulin was DRAQ5. For images where we stained with BODIPY TRc and anti-tubulin, we started by using SYBR Green I as a nuclear stain in these parasites (along with anti-tubulin/anti-mouse Alexa Fluor 647). As noted in the text, however, we found that SYBR Green I resulted in significant and rapid bleaching, so this largely precluded imaging using SYBR Green I in later parasite stages that can take up to 35 minutes to acquire each image. For this reason, we switched to SYTOX Red (along with anti-tubulin/anti-moues Alexa Fluor 488), which was much more photostable than SYBR Green I and effectively allowed us to image large cells with 4 markers, such as the schizonts in Supplementary Figure 7 (previously supplementary Figure 5), without significant bleaching. We reasoned that trying to include all of these nuclear stains, along with our experience using them, in this study would be something that would help others in selecting the right stain for their U-ExM experiments in future. In the interest of clarity, we also provided Supplementary Table 1, which for each image included in this study says exactly what each parasite depicted in our representative images were stained with, along with anything these images were stained with that is not shown in the representative image.
- BODIPY TRc should be referred to as a stain rather than marker.
We agree that BODIPY TRc is a stain and not a marker. This has been changed.
- Lines 449-452: (…) microtubules which extend from t=apical polar ring 2 … to the basal complex (Fig 1b) à This statement does not seem to be based on data shown in Fig 1b. Could the authors please rectify that?
The way this sentence was written conflated work that had been shown by others with what we see in Figure 1b and so the sentence has been rewritten to clarify this. It has been previously shown that subpellicular microtubules extend from apical polar ring 2 (Hanssen et al., 2013 (see inset Figure 1d)), and that in merozoites form fully segmented schizonts, the basal complex forms a ring at the very basal end of the merozoite (Rudlaff et al., 2020 (see inset figure 2a)). What our images show is the first combination of these two things together. Based on NHS staining, we can see an apical polar ring (marked APRs in Figure 1c), although it isn’t clear currently if this represents just one, or both APRs. Notably, the apical polar rings were also observed by NHS ester on U-ExM expanded P. berghei merozoites (Bertiaux et al., 2021(see inset figure 3)) Additionally, by NHS ester staining we see a dense ring at the back of the merozoite (marked BC in Figure 1c) that looks remarkably similar to the basal complex identified by FIB-SEM previously (Rudlaff et al., 2020 (see inset figure 2a)). In conjunction with the tubulin staining, we can see that the subpellicular microtubules extend from the APRs along the side of the merozoite, up until the basal complex. To the best of our knowledge, this is the first time that the APRs, subpellicular microtubules, and basal complex have all been seen together in a single image. Although there are many merozoites diagrams that have inferred this structure previously (Bannister et al., 2003 (see inset figure 4)).
This section of the results has been revised to make this more clear (line 288 – 300):
“At the apex of the rhoptry neck a ring structure can be observed (Figure 1b & c), which we inferred to be the apical polar rings based on its similar appearance to the apical polar rings in electron microscopy studies [37]. It is not clear if what we observe by NHS ester staining represents apical polar ring 1, apical polar ring 2, or both. At the basal end of the parasite, we observed another ring by NHS ester staining that is likely the basal complex (Figure 1b & c) based on its similarity to the basal complex as identified by FIB-SEM [12]. By combining NHS Ester and tubulin staining, we observed that subpellicular microtubules extend from the apical polar rings, along the length of the merozoite and end at the basal complex (Figure 1b &c). While previously published models of merozoites have speculated on this organization previously [38], to the best of our knowledge, this is the first time the apical polar rings, subpellicular microtubules, and basal complex have been observed in the same merozoite.”
Inset Figure 1 from Hanssen, E.; Dekiwadia, C.; Riglar, D.T.; Rug, M.; Lemgruber, L.; Cowman, A.F.; Cyrklaff, M.; Kudryashev, M.; Frischknecht, F.; Baum, J.; et al. Electron tomography of Plasmodium falciparum merozoites reveals core cellular events that underpin erythrocyte invasion. Cellular Microbiology 2013, 15, 1457-1472, doi:https://doi.org/10.1111/cmi.12132.
Inset Figure 2 from Rudlaff, R.M.; Kraemer, S.; Marshman, J.; Dvorin, J.D. Three-dimensional ultrastructure of Plasmodium falciparum throughout cytokinesis. PLOS Pathogens 2020, 16, e1008587, doi:10.1371/journal.ppat.1008587.
Inset Figure 3 from Bertiaux, E.; Balestra, A.C.; Bournonville, L.; Louvel, V.; Maco, B.; Soldati-Favre, D.; Brochet, M.; Guichard, P.; Hamel, V. Expansion microscopy provides new insights into the cytoskeleton of malaria parasites including the conservation of a conoid. PLOS Biology 2021, 19, e3001020, doi:10.1371/journal.pbio.3001020.
Inset Figure 4 from Bannister, L.H.; Hopkins, J.M.; Dluzewski, A.R.; Margos, G.; Williams, I.T.; Blackman, M.J.; Kocken, C.H.; Thomas, A.W.; Mitchell, G.H. Plasmodium falciparum apical membrane antigen 1 (PfAMA-1) is translocated within micronemes along subpellicular microtubules during merozoite development. Journal of Cell Science 2003, 116, 3825-3834, doi:10.1242/jcs.00665.
- Some of the figure (for example sup fig 5) were ha4rd to read – the authors should label structures and indicate clearly the structures that they refer to in the text (eg arrows, asterisks,…)
We agree with the author that some of the features were difficult to see in some images, particularly Figure 1 and Supplementary Figure 7 (previously Supplementary Figure 5). To both of these images we have added schematics to help for interpretation to the reader. We preference these over simply arrowheads or markers because U-ExM is new to Plasmodium, and even if a structure is pointed out, it may be difficult to see how we are interpreting that. The figure legends of the modified figures now read:
“Figure 1: Comparison between microtubule structures visualized in unexpanded and U-ExM P. falciparum asexual blood-stage parasites.
MCMBPHADD parasites, cultured in the presence of Shld1, were imaged using super-resolution Airyscan microscopy after being prepared for regular immunofluorescence assay (a), or U-ExM (b). All parasites were stained with a nuclear stain (Hoechst, DAPI, DRAQ5, or SYBR in cyan), anti-tubulin (in magenta) and a protein stain (N-hydroxysuccinimide (NHS) Ester in greyscale). All previously identified blood-stage microtubule structures (hemispindle, mitotic spindle, interpolar spindle and subpellicular microtubules) were observed by both IFA and U-ExM. Images in (a) represent a single z-slice from a z-stack image, while images in (b) are maximum-intensity projections. Slice-by-slice videos of images in 1b found in Supplementary Videos 1-4. Scale bars as labelled in each image, solid bars = XY scale, dashed bar = combined depth of slices used for Z-projection. (c) Expanded and annotated view of NHS Ester channel from mitotic spindle and subpellicular microtubule images from (b) along with schematic interpretation of these images. Arrowheads point to NHS staining of interest. Colors in schematic: black = dense NHS Ester staining, grey = light NHS Ester staining, blue = DNA, purple = microtubules. RBC = Red blood cell membrane, PVM = Parasitophorous vacuole membrane, PPM = Parasite plasma membrane, MTOC = Microtubule organizing center, MTs = microtubules, APRs = Apical polar rings, RN = Rhoptry neck, RB = Rhoptry bulb, BC = Basal complex, Rh = rhoptry, SPMTs = Subpellicular microtubules.”
“Supplementary Figure 7: MCMBP deficient parasites display subpellicular microtubules and complete improper segmentation.
(a) MCMBPHADD parasites were cultured either in the presence, or absence, of Shld1 and schizonts were arrested before egress using E64. Parasites were prepared for U-ExM, stained with a nuclear stain (SYTOX, in cyan), anti-tubulin (in magenta), a membrane stain (BODIPY TRc, in white), and a protein stain (NHS Ester, in greyscale, and visualized using Airyscan microscopy. Note that -Shld1 image rows 2 and 3 are zoomed in merozoites from the image in row 1. Images containing BODIPY TRc are average intensity projections, while those with NHS ester are maximum intensity projections. Slice-by-slice videos of images can be found in Supplementary Videos 20-21. Scale bars as labelled in each image, solid bars = XY scale, dashed bar = combined depth of slices used for Z-projection. (b) Schematic representations of large and zoid merozoites from (a). Rh = Rhoptries, SPMTs = Subpellicular microtubules, APRs = Apical polar rings Black and red dashed images are the images represented in (b).”
- Line 312. Remove “due to”
This has been fixed.
- Line 550. “data no shown”; not
This has been fixed.
- Line 338 “parasites cultures”; cultured
This has been fixed.
